# Natural History of Hyperphagia in Patients with Pseudohypoparathyroidism

**DOI:** 10.3390/jcm14155345

**Published:** 2025-07-29

**Authors:** Jaclyn Tamaroff, Ashley H. Shoemaker

**Affiliations:** Division of Pediatric Endocrinology and Diabetes, Department of Pediatrics, Vanderbilt University Medical Center, Nashville, TN 37212, USA; jaclyn.tamaroff@vumc.org

**Keywords:** pseudohypoparathyroidism, PHP1A, PHP1B, pseudopseudohypoparathyroidism, obesity, hyperphagia

## Abstract

**Background/Objectives**: Pseudohypoparathyroidism (PHP) is a group of genetic disorders characterized by end-organ resistance to multiple hormones, short stature, brachydactyly, subcutaneous ossifications, obesity, and developmental delays. The tissue specific imprinting of *GNAS* in the hypothalamus may lead to different eating behavior phenotypes in maternally inherited (PHP1A, PHP1B) vs. paternally inherited (PPHP) variants. In this exploratory study, we aimed to evaluate differences in eating behaviors in a cohort of patients with PHP1A, PPHP and PHP1B. **Methods**: Assessments included caregiver-reported measures (hyperphagia questionnaire, children’s eating behavior questionnaire, child feeding questionnaire) and self-reported measures (three factor eating behavior questionnaire). **Results**: A total of 58 patients with PHP1A, 13 patients with PPHP and 10 patients with PHP1B contributed data, along with 124 obese pediatric controls. An increased risk of obesity was found in PHP1A vs. PPHP (adult body mass index (BMI) 39.8 ± 8.7 vs. 30.2 ± 7.4 kg/m^2^, *p* = 0.03). Parents reported significantly earlier onset of interest in food in children with PHP1A (2.0 ± 2.3 years) and PHP1B (1.1 ± 1.3 years) compared with controls (5.2 ± 3.2 years, *p <* 0.001). Measures of hyperphagia, satiety and other feeding behaviors were all similar to controls. The highest hyperphagia questionnaire scores were seen prior to adolescence. In a multi-year, longitudinal assessment of 11 pediatric patients with PHP1A, hyperphagia scores were stable and 25% showed an improvement in symptoms. **Conclusion**: Patients with PHP1A/1B may have hyperphagia symptoms from a young age but they do not worsen over time. Patients may overeat when allowed access to food, but do not usually have disruptive food seeking behaviors. Early diagnosis can give clinicians the opportunity to provide anticipatory diagnosis on the increased risk of obesity in PHP1A/1B and need for scheduled meals and controlled portions. Further studies with larger cohorts are needed to confirm these findings.

## 1. Introduction

Pseudohypoparathyroidism (PHP) is a group of genetic disorders characterized by end-organ resistance to parathyroid hormone (PTH) and often the Albright hereditary osteodystrophy phenotype (AHO; short stature, brachydactyly, subcutaneous ossifications, obesity and developmental delays) with a prevalence of 0.3–1.2 cases per 100,000 population [1,2,3,4,5]. The classic form of the disorder is PHP1A, caused by the maternally inherited loss of function variants in the gene *GNAS*. Most patients with PHP1A will have multiple-hormone resistance due to abnormal function of the *GNAS*-encoded stimulatory G-protein alpha subunit which is utilized by several G-protein-coupled hormone receptors. *GNAS* is an imprinted gene with preferential expression of the maternal allele in some tissues, including the thyroid, kidney, hypothalamus, and pituitary gland. Due to this imprinting phenomenon, paternally inherited loss of function *GNAS* variants do not result in hormone resistance and the resulting phenotype is called pseudopseudohypoparathyroidism (PPHP). Alternately, abnormal *GNAS* function can be caused by methylation defects at the *GNAS* differentially methylated regions (including the gene *STX16*) or paternal 20q disomy. These methylation defects are called PHP1B and have a variable phenotype, from isolated PTH resistance to a PHP1A-like phenotype. There is significant clinical overlap between the various forms of PHP.

A European group has recommended a different nomenclature, inactivating PTH/PTHrP signaling disorder (iPPSD) [6]. Loss of function variants in *GNAS* are classified as iPPSD2 and methylation changes in *GNAS* are classified as iPPSD3. Tissue specific imprinting of *GNAS* in the hypothalamus may lead to different eating behavior phenotypes in maternally inherited (PHP1A) vs. paternally inherited (PPHP) variants, and a previous study showed that patients with PHP1A have significantly higher BMI than patients with PPHP [7]. For the purposes of this study, we will use the PHP nomenclature (PHP1A, PPHP) as it distinguishes between maternally and paternally inherited variants rather than grouping them into one iPPSD category.

We have previously published two small studies evaluating eating behaviors in PHP1A. The first study included questionnaire data from 10 patients with PHP1A, while the second study included questionnaire and buffet meal data from 16 patients with PHP1A and 3 patients with PHP1B [8,9]. Patients with PHP1A had an increased interest in food at a younger age compared to controls, but only a subset of patients had significant hyperphagia. It is not clear if there are other differences between the groups with and without hyperphagia. There are no reports on eating behaviors of patients with PPHP.

In this exploratory study, we aimed to evaluate differences in eating behaviors in a cohort of patients with PHP1A, PPHP and PHP1B. We also compared behavior over time to test the hypothesis that patients with PHP1A have early but non-progressive hyperphagia.

## 2. Materials and Methods

Data collection was approved by the Vanderbilt Institutional Review Board (IRB: 170219, approved 13 March 2017). Informed consent was obtained from all adult subjects. For pediatric participants, consent was obtained from their legal guardian and assent was obtained from the participant.

Cross-sectional data was collected through online surveys and from baseline data in patients enrolled in clinical trials (NCT04551170 and NCT03029429). The longitudinal analysis includes previously published data (PMID 25337124 and PMID 30085125). Pediatric control patient data was collected through the Vanderbilt Childhood Obesity Registry (NCT02957916) which enrolled children with onset of obesity prior to 10 years old. We also included control patient data from the previously published studies (PMID 25337124 and PMID 30085125), if the subject had a BMI > 95th percentile for age and gender.

All participants met clinical criteria for diagnosis of PHP. Clinical classification was based on their medical records and self-report. Not all patients had genetic testing available for review. We had genetic confirmation for diagnosis in 62.1% of individuals with PHP1A and 40% of individuals with PHP1B. All patients with PPHP had a genetic diagnosis.

Surveys were administered via REDCap (Research Electronic Data Capture) [10]. REDCap is a secure, web-based application designed to support data capture for research studies. Height and weight were obtained from the medical record or patient report if records were unavailable (adults only). Pediatric z-scores were calculated as standard deviations from the mean using gender and age specific Centers for Disease Control growth charts. All children had a Hyperphagia Questionnaire (HQ) [11], Children’s Eating Behavior Questionnaire (CEBQ) [12], and Child Feeding Questionnaire (CFQ) [13] completed by the primary caregiver. Adults completed the Three Factor Eating Questionnaire (TFEQ) [14]. Adults enrolled in the clinical trial also completed the HQ.

The HQ was originally developed to assess hyperphagia in Prader–Willi syndrome and contains 11-questions that assess symptoms of hyperphagia in one of three categories (Behavior, Drive and Severity. The 35-item CEBQ assesses Positive and Negative Eating Behaviors. The Positive Eating Behavior score comprises four sub-scales: Food Responsiveness, Enjoyment of Food, Emotional Overeating and Desire to Drink. The Negative Eating Behavior score is composed of four sub-scales: Satiety Responsiveness, Slowness in Eating, Emotional Undereating and Food Fussiness. The CFQ assesses parental beliefs, attitudes and practices around child feeding. There are seven factors in the CFQ model, Perceived responsibility, Perceived parent weight, Perceived child weight, Concern about child weight, Restriction, Pressure to eat, and Monitoring. The TFEQ contains 51 questions to assess Restraint, Disinhibition and Hunger in adults.

Analysis: Analyses were performed using SPSS software, version 29. Significance was set at a two-sided *p*-value < 0.05. Groups were compared using Kruskal–Wallis test. If this test was significant, each group was compared using the Mann–Whitney U test, adjusted by the Bonferroni correction. For categorical variables, a Chi Squared test was used for comparison.

## 3. Results

### 3.1. Participant Characteristics

A total of 58 patients with PHP1A, 13 patients with PPHP and 10 patients with PHP1B contributed data (Table 1), along with 124 obese pediatric controls. Baseline data for the pediatric and adult cohorts are presented in Table 2 and Table 3. As expected, patients with PHP1A and PPHP have below average height. The degree of obesity was more severe in the control group than the PHP patients. The adult PPHP group was older and female as most were diagnosed later in life after having a child with PHP1A. Two out of four (50%) children with PHP1B were obese, versus 33 out of 35 (94%) children with PHP1A.

### 3.2. Hyperphagia Questionnaires

#### 3.2.1. Pediatric Hyperphagia

Results from the parent-completed HQ, CEBQ, and CFQ are presented in Table 4.

Parents reported significantly earlier onset of interest in food in children with PHP1A (2.0 ± 2.3 years) and PHP1B (1.1 ± 1.3 years) compared with controls (5.2 ± 3.2 years, *p* < 0.001). The effect size (r) for age of onset of hyperphagia between those with PHP1A and obese controls was r = −0.40 with a 95% confidence interval for the difference between medians of −4.0 to −2.0 years. For those with PHP1B and obese controls, r = −0.25 (95% confidence interval −7.0 to −1.0 years).

On the CFQ, parents of children with PHP1A also reported greater concern about their child’s weight (4.2 ± 0.8 vs. 3.3 ± 0.4. *p* < 0.001, score range 1–5). The effect size for perceived weight between individuals with PHP1A and obese controls was r = 0.59 with a 95% confidence interval for the difference between medians of 0.63 to 1.4. Parents of children with PHP1A also reported greater concern regarding weight than those with children with PPHP. For those with PHP1A compared to PPHP r = 0.41 with a 95% confidence interval of 0.25 to 4.0.

Measures of hyperphagia, satiety and other feeding behaviors were all similar between PHP1A and controls.

There was no difference in HQ Total score, CEBQ Positive Eating Behaviors or CEBQ Negative Eating Behaviors between PHP1A and obese controls after adjusting for age, gender and BMI in a linear regression model. Higher BMI, however, was associated with higher HQ Total score (β = 0.24, *p* = 0.003) and CEBQ Positive Eating Behaviors (β = 0.24, *p* = 0.007) and lower Negative Eating Behaviors (β = −0.18, *p* = 0.05).

While some patients with PHP1A have high levels of hyperphagia, there was significant variability and no evidence of worsening hyperphagia with age. The highest HQ scores were seen prior to adolescence (Figure 1). In 11 pediatric patients with multiple measurements over time, most patients had stable HQ scores and 3 patients (25%) showed improvement in symptoms with more than a 10-point decrease in total score (Figure 2).

#### 3.2.2. Adult Hyperphagia

Adults completed the TFEQ and results are presented in Table 5. There were no significant differences between groups. Results were similar baseline measurements from other groups with obesity [15].

## 4. Discussion

In a cohort of patients with PHP1A, we found the early onset of increased interest in food at an average of 2 years old. Patients with PHP1B showed a similar early onset of increased interest in food, though there were only 4 participants with available data. This correlates with findings early onset obesity, typically beginning in the first 1–2 years of life, in both PHP1A and PHP1B [16,17,18]. Comparatively, obese controls had increased interest in food at an average age of 5 years. Clinically, when a family reports very early onset of hyperphagia, this may be an important indicator for the need for genetic or biochemical testing. Additionally, earlier age of hyperphagia onset may indicate that earlier discussions on healthy eating patterns could be beneficial. Individuals with PHP1A are at increased risk of decreased insulin sensitivity, hyperglycemia, and diabetes, and understanding the underpinnings of their obesity is critical to help prevent and manage these metabolic complications [19].

While the PHP1B group did include adult and pediatric patients with obesity, the prevalence was lower and the PHP1B group also had non-significantly lower scores on measures of hyperphagia. Phenotypic overlap between PHP1A and PHP1B is increasingly recognized, including the presence of early-onset obesity, but a higher percentage of patients with PHP1A have the classic AHO physical features [20].

Early-onset obesity in PHP1A is attributed to abnormal function of the melanocortin-4 receptor in the hypothalamus, an area affected by tissue specific imprinting [16,21]. Due to the presence of paternal imprinting of *GNAS*, patients with PPHP (paternal inheritance) are not expected to have hypothalamic disruption of energy balance. Accordingly, we found lower rates of obesity in adults and children with PPHP, consistent with the previously published study by Long et al. [7]. Despite the lower BMI of the PPHP cohort, the adult PPHP group did not report lower levels of hunger on the TFEQ. The two pediatric PPHP patients did have lower hyperphagia and hunger scores, but a larger cohort is needed. It is also possible that the difference in weight gain in PHP1A vs. PPHP is driven more by differences in basal metabolic rate rather than food intake, but further research is needed on patients with PPHP [9,22].

The parent-reported degree of hyperphagia in PHP1A (mean total HQ score 24.2 ± 8.4) was less severe than that seen in patients with Prader–Willi syndrome (30.47 ± 4.52), but closer to other causes of syndromic obesity such as Bardet–Biedl syndrome (27.6 ± 9.0) and BDNF haploinsufficiency (26.37 ± 7.32) [11,23,24]. Differences between the PHP groups and controls in our study may have been blunted due to the severity of obesity in our control group, as hyperphagia increased with increasing BMI. Previous obese control groups in studies from our group and others have had HQ total scores ~19 vs. 23.2 in this study [9,23,24]. Of note, the HQ version used in this study has more questions (13 vs. 9) and a different scoring range than the Hyperphagia Questionnaire for Clinical Trials that is currently being used in industry trials so those data cannot be compared [25].

This exploratory study has multiple limitations, particularly related to sample size. Though this study included a large number of children with PHP1A, the limited sample size in the PHP1B, PPHP, and longitudinal cohorts limits the generalizability of the study. Generalizability may be further limited by selection bias from convenience sampling and as not all participants completed each survey. Though PHP is primarily a clinical diagnosis [26], an additional potential limitation is related to diagnostic uncertainty as some patients diagnosis relied on medical records and self-report rather than genetic testing. As with any survey studies, recall bias presents an additional concern, particularly related to remote timing of hyperphagia onset. Finally, multiple comparisons were tested which could increase the risk of type 1 errors, though mitigated by use of the Bonferroni correction. Further research is needed to confirm the findings of this exploratory study.

This is the first study to look at longitudinal data on hyperphagia in PHP. Patients with PHP1A may have hyperphagia symptoms from a young age but they do not worsen over time. This is different from Prader–Willi syndrome where hyperphagia develops over time with increasing severity throughout childhood. Though the longitudinal cohort was small, some patients even showed a resolution of symptoms as they aged. Based on this study and our clinical experience with patients, we suspect that people with PHP1A and possibly PHP1B may overeat when allowed access to food, but they do not usually have disruptive food seeking behaviors (HQ behavior subscale). Early diagnosis gives clinicians the opportunity to provide anticipatory diagnosis on the increased risk of obesity and need for scheduled meals and controlled portions. These strategies are helpful in patients with Prader–Willi syndrome, despite much more severe hyperphagic behaviors. The challenge with PHP is to make the diagnosis before obesity becomes severe. Unfortunately, obesity is still one of the main presenting features of this genetic disorder.

## Figures and Tables

**Figure 1 jcm-14-05345-f001:**
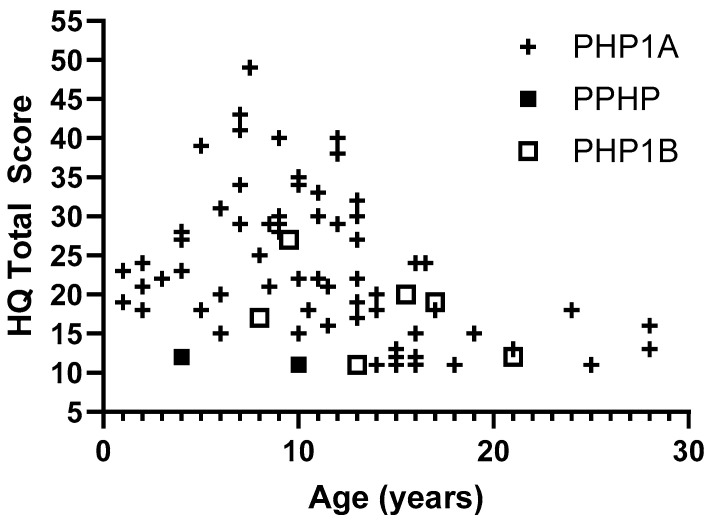
Hyperphagia questionnaire total scores by age.

**Figure 2 jcm-14-05345-f002:**
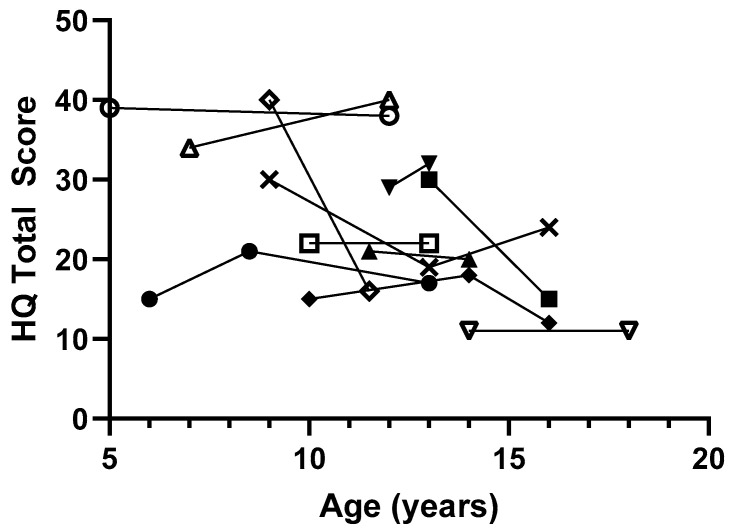
Hyperphagia questionnaire total scores over time for 11 pediatric subjects with PHP1A. Each symbol represents a different subject.

**Table 1 jcm-14-05345-t001:** Classification of pseudohypoparathyroidism.

	Genetics	Major Symptoms	Study Population
PHP1A (iPPSD2)	Maternally inherited LOF variant in *GNAS*	- Multiple-hormone resistance -AHO phenotype - Early onset obesity	Pediatric: 44 Adult: 14
PPHP (iPPSD2)	Paternally inherited LOF variant in *GNAS*	-Short stature -AHO phenotype (without obesity)	Pediatric: 2 Adult: 11
PHP1B (iPPSD3)	Methylation defects or paternal 20q disomy	-Variable: isolated PTH resistance to PHP1A-like	Pediatric: 6 Adult: 4

PHP: pseudohypoparathyroidism, PPHP: pseudopseudohypoparathyroidism, iPPSD: inactivating PHT/PTHrP signaling disorder, LOF: loss of function, AHO: Albright hereditary osteodystrophy phenotype (short stature, obesity, brachydactyly, subcutaneous ossifications).

**Table 2 jcm-14-05345-t002:** Baseline characteristics of the pediatric cohort.

	PHP1A (n = 44)	PPHP (n = 2)	PHP1B (n = 6)	Obese Controls (n = 124)	*p*-Value
Gender (% female)	65.9	100	50.0	73.4	0.39
Age (years)	9.7 ± 4.8	7.0 ± 4.2	10.8 ± 4.6	10.1 ± 3.6	0.68
Height Z-score	−0.41 ± 1.55 * (n = 35)	−0.75 (n = 1)	0.83 ± 1.02 (n = 4)	1.35 ± 1.32 *	**<0.001**
Weight Z-score	1.96 ± 0.99 * (n = 35)	−0.79 (n = 1)	1.47 ± 1.16 (n = 4)	2.95 ± 0.98 *	**<0.001**
BMI Z-score	2.27 ± 0.66 * (n = 35)	−0.21 (n = 1)	1.33 ± 1.12 * (n = 4)	2.60 ± 0.57 *	**<0.001**
BMI (% of the 95th percentile)	127 ± 28 * (n = 35)	84 (n = 1)	104 ± 30 (n = 4)	150 ± 37 *	**<0.001**

PHP1A, pseudohypoparathyroidism type 1A; PPHP, pseudohypoparathyroidism; PHP1B, pseudohypoparathyroidism type 1B. Overall *p*-value by Kruskall–Wallis Test or Chi Squared Test. * Designates differences between pairs by Mann–Whitney U Test, adjusted by Bonferroni correction with significant *p*-values in bold Height and weight z-scores were significantly higher in obese controls vs. PHP1A (both *p* < 0.001). BMI z-scores were significantly higher in obese controls vs. both PHP1A (*p* = 0.02) and PHP1B (*p* = 0.03). The BMI percent of the 95th percentile (BMI95) was significantly higher in obese controls vs. PHP1A (*p* < 0.001).

**Table 3 jcm-14-05345-t003:** Baseline characteristics of the adult cohort.

	PHP1A (n = 14)	PPHP (n = 11)	PHP1B (n = 4)	*p*-Value
Gender (% female)	78.6	100	66.7	0.19
Age (years)	27.9 ± 10.6	36.9 ± 10.0	33.0 ± 9.5	0.09
Height (cm)	151.7 ± 8.2 (n = 8)	154.6 ± 4.9 (n = 7)	173.4 (n = 1)	0.24
Height Z-score	−1.79 ± 1.24 (n = 8)	−1.35 ± 0.75 (n = 7)	−0.48 (n = 1)	0.34
Weight (kg)	92.4 ± 24.9 (n = 8)	72.3 ±19.2 (n = 7)	132.3 (n = 1)	0.08
BMI (kg/m^2^)	39.8 ± 8.7 (n = 8)	30.2 ± 7.4 (n = 7)	44.0 (n = 1)	0.05

PHP1A, pseudohypoparathyroidism type 1A; PPHP, pseudohypoparathyroidism; PHP1B, pseudohypoparathyroidism type 1B. Overall *p*-value by Kruskall–Wallis Test or Chi Squared Test.

**Table 4 jcm-14-05345-t004:** Questionnaire results for the pediatric cohort.

	PHP1A	PPHP	PHP1B	Obese Controls	*p*-Value
*Hyperphagia Questionnaire*	(n = 43)	(n = 2)	(n = 6)	(n = 123)	
Behavior	9.4 ± 3.8	5.0 ± 0	7.7 ± 3.4	9.9 ± 3.7	0.05
Drive	10.7 ± 3.9	4.5 ± 0.7	7.3 ± 2.3	9.8 ± 3.8	0.03
Severity	4.1 ± 1.9	2.0 ± 0	3.0 ± 1.1	3.5 ±1.8	0.10
Total	24.2 ± 8.4	11.5 ± 0.7	18.0 ± 16.0	23.2 ± 8.4	0.04
Age of hyperphagia onset (years)	2.0 ± 2.3 * (n = 37)	n/a	1.1 ± 1.3 * (n = 4)	5.2 ± 3.2 * (n = 105)	**<0.001**
*Child Eating Behavior Questionnaire*	(n = 42)	(n = 2)	(n = 6)	(n = 98)	
Positive Feeding Behaviors	52.3 ± 11.6	32.0 ± 0	47.0 ± 16.9	54.3 ± 12.1 (n = 95)	0.08
Food Responsiveness	16.2 ± 5.6	8.0 ± 1.4	15.3 ± 7.0	16.8 ± 5.2	0.17
Emotional Overeating	9.7 ± 3.5	5.0 ± 1.4	8.8 ± 3.8	11.2 ± 4.0 (n = 97)	0.02
Enjoyment of Food	16.6 ±5.6	14.5 ± 3.5	13.7 ± 6.4	16.3 ± 3.3	0.49
Desire to Drink	9.7 ± 3.5	4.5 ± 0.7	9.2 ± 3.7	9.8 ± 3.2 (n = 97)	0.18
Negative Feeding Behaviors	50.9 ± 8.5 (n = 41)	46.0 ± 2.8	56.4 ± 11.5 (n = 5)	49.4 ± 10.6 (n = 94)	0.44
Satiety Responsiveness	12.9 ± 3.4	18.5 ± 0.7	16 ± 4.9	11.9 ± 3.9 (n = 95)	0.03
Slowness in Eating	10.2 ± 2.9	11.0 ± 1.4	10.4 ±3.0 (n = 5)	9.7 ± 3.4	0.72
Emotional Undereating	10.8 ± 3.5 (n = 41)	5.5 ± 0.7	10.7 ± 4.8	10.7 ± 2.8	0.16
Fussiness	17.1 ± 4.9	11.0 ± 4.2	23.3 ± 7.1	16.9 ± 5.8	0.06
*Child Feeding Questionnaire*	(n = 24)	(n = 2)	(n = 5)	(n = 17)	
Perceived responsibility	4.2 ± 0.7	4.0 ± 0	3.9 ± 0.9	3.7 ± 0.8	0.15
Perceived parent weight	3.3 ± 0.8 (n = 23)	3.1 ± 0.2	3.2 ± 0.1	3.5 ± 0.6	0.52
Perceived child weight	4.2 ± 0.8 *	1.9 ± 1.3 *	3.5 ± 0.6	3.3 ± 0.4 *	**<0.001**
Concern about child weight	3.8 ± 1.2	1.0 ± 0	2.7 ±1.6	3.6 ± 1.3	0.06
Restriction	3.6 ± 0.7	2 ± 0.2	3.4 ± 1.8	3.8 ± 0.9	0.19
Pressure to eat	2.2 ± 1.4	2.1 ± 1.6	2.9 ± 0.5	2.1 ± 0.9	0.27
Monitoring	4.0 ± 1.0	2.5 ± 2.1	2.9 ± 0.9	4.0 ± 0.7	0.10

PHP1A, pseudohypoparathyroidism type 1A; PPHP, pseudohypoparathyroidism; PHP1B, pseudohypoparathyroidism type 1B. Overall *p*-value by Kruskall–Wallis Test. * Designates significant differences between pairs by Mann–Whitney U Test, adjusted by Bonferroni correction with significant *p*-values in bold. Age of hyperphagia onset was significantly younger in PHP1A vs. obese controls (*p* < 0.001) and PHP1B vs. obese controls (*p* = 0.02). As those with PPHP did not have significant hyperphagia, the age of onset was not applicable. Perceived child weight scores were significantly higher in PHP1A vs. both obese controls (*p* < 0.001) and PPHP (*p* = 0.02).

**Table 5 jcm-14-05345-t005:** Questionairre results for the adult cohort.

	PHP1A	PPHP	PHP1B	*p*-Value
*Three Factor Eating Questionnaire*	(n = 12)	(n = 11)	(n = 3)	
Restraint	8.6 ± 4.1	10.6 ± 5.0	7.3 ±2.1	0.28
Disinhibition	5.2 ± 3.7	6.5 ± 3.8	7.0 ± 2.8	0.44
Hunger	4.8 ± 4.2	5.6 ± 4.4	4.0 ± 5.2	0.60

PHP1A, pseudohypoparathyroidism type 1A; PPHP, pseudohypoparathyroidism; PHP1B, pseudohypoparathyroidism type 1B. Overall *p*-value by Kruskall–Wallis Test.

## Data Availability

The data presented in this study are available on request from the corresponding author due to subject privacy concerns due to rare disease and small sample size.

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
