# Peer review of "Natural History of Hyperphagia in Patients with Pseudohypoparathyroidism"

_jcm, 2025, doi:10.3390/jcm14155345_

Round 1

Reviewer 1 Report

Comments and Suggestions for Authors

OVERALL ASSESSMENT This manuscript addresses an important clinical question regarding eating behaviors in patients with pseudohypoparathyroidism (PHP), a rare genetic disorder. The authors present cross-sectional and longitudinal data on hyperphagia patterns across different PHP subtypes. While the research question is clinically relevant and the longitudinal component provides novel insights, the study has significant methodological limitations that substantially impact the reliability and generalizability of the findings.  

  1. Methodology and Study Design

Critical Issues: Statistical Error (Line 105): The manuscript states "Significance was set at a two-sided p-value <0.5" which appears to be a typographical error. This should read p<0.05. This fundamental error raises concerns about the statistical rigor of the analysis. Sample Size Limitations: The study is severely underpowered for meaningful comparisons:

  • PPHP pediatric group: n=2 (inadequate for any statistical analysis)
  • PHP1B groups: n=6 pediatric, n=4 adult (very limited)
  • Longitudinal analysis: only 12 patients
  • No power calculations provided

Diagnostic Accuracy: The authors acknowledge that "Not all patients with PHP1A/1B had genetic testing available for review" and relied on "clinical classification based on their medical records and self-report." This introduces significant diagnostic uncertainty that could affect the validity of group comparisons. Recommendations:

  • Correct the statistical significance threshold
  • Acknowledge the severe limitations imposed by small sample sizes
  • Provide power calculations or acknowledge the study as exploratory
  • Discuss the impact of diagnostic uncertainty on findings
  1. Results and Data Analysis

Concerns: Misleading Sample Description: The abstract and text refer to a "large cohort," which is inappropriate given the small PPHP (n=13) and PHP1B (n=10) groups. Multiple Comparisons: Extensive statistical testing without adequate correction increases the risk of Type I errors. Missing Data: Inconsistent sample sizes across analyses (varying n values in tables) are not systematically addressed. Clinical vs. Statistical Significance: The manuscript does not adequately distinguish between statistical significance and clinical relevance. Key Findings Assessment:

  • Early food interest onset: This is the most robust finding and clinically meaningful
  • Non-progressive hyperphagia: Interesting but based on limited longitudinal data
  • Group differences: Many comparisons are underpowered and should be interpreted cautiously

Recommendations:

  • Remove references to "large cohort"
  • Provide confidence intervals and effect sizes for key findings
  • Acknowledge limitations of multiple comparisons
  • Discuss clinical significance of findings more thoroughly
  1. Discussion and Conclusions

Major Deficiencies: Inadequate Limitations Section: The manuscript lacks a proper discussion of study limitations, which is essential given the methodological concerns. Overstated Conclusions: Claims about clinical management recommendations are not well-supported by the limited data. Generalizability: The findings may not be generalizable given the convenience sampling and small sample sizes. Required Additions:

  • Comprehensive limitations section addressing:
    • Small sample sizes and statistical power
    • Diagnostic uncertainty
    • Selection bias from convenience sampling
    • Recall bias for hyperphagia onset timing
  • More conservative interpretation of findings
  • Discussion of future research needs

This manuscript addresses an important clinical question and provides some valuable insights into eating behaviors in PHP. However, significant revisions are required before it meets publication standards for the Journal of Clinical Medicine. Priority Actions:

  1. Correct the statistical error regarding p-value threshold
  2. Add a comprehensive limitations section
  3. Revise abstract and conclusions to be more conservative
  4. Acknowledge the exploratory nature of findings given sample size limitations
  5. Improve figure quality and data presentation

Secondary Actions:

  1. Standardize nomenclature throughout
  2. Improve reference formatting
  3. Enhance discussion of clinical implications with appropriate caveats

The authors should consider this as an exploratory study that provides preliminary evidence requiring replication in larger, well-powered studies. With appropriate revisions acknowledging these limitations, the manuscript could make a valuable contribution to the literature on this rare disorder.

Reviewer 2 Report

Comments and Suggestions for Authors

In this original article authors conducted a longitudinal case-control study about eating habits between patients affected by different types of pseudohypoparathyroidism and a pediatric population with obesity. Moreover they considered pseudohypoparathyroidism’s features also in adult setting.

I find the subject of extreme interest because of the lack of reliable scientific, as well underlined by the authors.

Nonetheless, there are some major issues that should be resolved:

  • In the introduction epidemiological data reported in lines 39-40 are slightly out-date. Please verify if any other data are available.
  • In “material and methods” section, line 81, you specified that not all patients with pseudohypoparathyroidism had genetic testing available: the precise number of patients with and without genetic test should be reported.
  • In “material and methods” section, line 106 you stated that a p-value <0,5 was considered significative: is it a typing error for 0,05? Please specified alfa-error that you considered in statistical analysis.

As minor issues:

  • I think it could be useful to underline ethical approval also in material and methods, not only as final statement.
  • If available it could be useful to add levels of glycemia and insulin and to compare them with referred population: this analysis, if not available, is not necessary. Anyway you could add a little section in discussion part regarding possible insulin resistance pathways in pseudohypoparathyroidism patients.
  • The acronyms should be introduced the first time in the text and then always used as substitute of the original word (example: AHO for Albright Hereditary Osteodystrophy).

Round 2

Reviewer 1 Report

Comments and Suggestions for Authors

The authors addressed all my previous suggestions. Therefore, in my view, the manuscript should be accepted. 

Reviewer 2 Report

Comments and Suggestions for Authors

Thanks to the authors for thoroughly and promptly addressing the previously raised comments. The paper has been significantly improved, and I believe it can now be considered ready for publication.